# New spectral-parameter dependent solutions of the Yang-Baxter equation

Alexander. S. Garkun[1], Suvendu K. Barik[2], Aleksey K. Fedorov[3,4,5] and Vladimir Gritsev[2,3*]

**1** Institute of Applied Physics of the National Academy of Sciences of Belarus, Belarus
**2** Institute of Physics, University of Amsterdam, The Netherlands
**3** Russian Quantum Center, Moscow, Russia
**4** National University of Science and Technology "MISIS", Moscow, Russia
**5** P. N. Lebedev Physical Institute of the Russian Academy of Sciences, Moscow, Russia
* V.Gritsev@uva.nl

January 29, 2024

## Abstract

**The Yang-Baxter Equation (YBE) plays a crucial role for studying integrable many-body quantum systems. Many known YBE solutions provide various examples ranging from quantum spin chains to superconducting systems. Models of solvable statistical mechanics and their avatars are also based on YBE. Therefore, new solutions of the YBE could be used to construct new interesting 1D quantum or 2D classical systems with many other far-reaching applications. In this work, we attempt to find (almost) exhaustive set of solutions for the YBE in the lowest dimensions corresponding to a two-qubit case. We develop an algorithm, which can potentially be used for generating new higher-dimensional solutions of the YBE.**

# 1   Introduction

Solvable and integrable many-body systems form a basis for our understanding of contemporary physics [1, 2]. They played a fundamental role in our knowledge of universality classes of equilibrium phase transitions in classical and quantum systems [3]; they establish certain universal concepts in low-dimensional physics, such as the Luttinger liquid, sine-Gordon and impurity systems, Hubbard model, to name a few [2]. They also play a crucial role in studies of out-of-equilibrium dynamics, for isolated and open quantum systems [4]. These models, while fine-tuned, are representatives of a much broader universality classes of low-dimensional physics behaviours. Therefore, expanding the arsenal of solvable models has far-promising potential for exploring new physics and mathematics. This calls for a systematic way of identifying solvable models in a broad sense. Here we focus on a subclass of solvable models known as Yang-Baxter (YB) integrable models. In a broader context YB-type models found extensive appearance in many physical systems ranging from condensed matter and cold atomic systems (for an overview, see e.g. Ref. [5]) to high-energy physics [6], AdS/CFT correspondence [7], gauge theory [8], quantum computing [9–11], and mathematics [12].

In the latter case, we note that there are deep connections between the concept of quantum integrability and quantum computing. First of all, a unified description of universal quantum computing models and the YBE has been introduced (to our knowledge) by Kauffmann and collaborators in Ref. [9] and developed later in Refs. [10,11]. Existing noisy intermediate-scale quantum (NISQ) computing devices can be used to study the transition between integrable and non-integrable regimes for complex many-body systems realised as quantum circuits [13–15]. Recently, such problem has been studied with the use of the 46-qubit superconducting processor for the 1D Heisenberg model [15]. At the same time, integrable models and corresponding integrable quantum circuits lead to the smaller degree of quantum scrambling during the course of the quantum computation if compared with the non-integrable case, which can be interesting for certain computing protocols and quantum error correction [9], leading to an efficient compression schemes [16]. The relation between the YBE and algebraic approaches are also important in topological quantum computations, where YBEs describe braiding of

anyons [17].

Thus, the task of obtaining a full classification of the YBE solutions is a vital problem for further developments in various fields; finding new solutions of the YBE opens a way to construct new integrable quantum many-body models with far-reaching applications. This is the purpose of quite an ambitious plan: analyzing the YBE in general settings which is, however, a challenging problem. Here we make steps in this direction and introduce a possible scheme to be developed further.

In general, finding a classification scheme for solutions to the YBE is a formidable task. To our knowledge the first attempt in this direction has been made by Kulish and Sklyanin [18] as well as later by Reshetikhin and Wiegman [19] and by Jimbo [20] using the tools of classical and quantum groups. However, all these solutions are based on special structures inherited from the classical or quantum Lie algebras. These "symmetric" (in a sense that they are related to certain Lie symmetries or their q-deformations) solutions have found numerous applications in condensed matter theory, field theory and string theory. Later on, Hietarinta provided a classification of a 4-dimensional representation of the braid group, namely the YBE with no spectral parametric dependence [21, 22].

Recently, Vieira [23] has provided a classification scheme for a number of 4-to 8 vertex models referring to some earlier studies [24,25]. These studies have been extended furhter see Refs. [26–30]; see also recent works by Pozsgay with coauthors with partial classifications [31] and concrete examples [32]. These models have a number of symmetries and properties which are listed carefully in the aforementioned manuscripts [1].

Here we relax these conditions and look into a more general class of models. For example, we relax the condition of regularity (i. e. nonzero determinant) of the $R$-matrix which solves the YBE. While this condition is important for Algebraic Bethe Ansatz approach for quantum 1D many-body systems or 2D statistical mechanics models, it is not necessary for quantum information applications or quantum circuits. In a sense we generalize above approaches and develop a systematic procedure for obtaining potentially any new solutions in any dimension for the spectral-parameter dependent YBE solutions. We explicitly present a large class of solutions for the lowest dimensions in terms of $4 \times 4$ matrices.

The paper is organized as follows. First we provide a list of (hopefully) new found solutions and then describe our procedure which can easily be implemented for searching new higher-dimensional YBE solutions. For illustrational purposes in the Appendix we list $R$-matrices, which are obtained by our procedure before implementing all symmetries and similarity transformation listed in Sec. 2.2.

## 2 Preliminaries

### 2.1 The Yang-Baxter Equation

The (quantum) YBE with an arbitrary set of spectral parameters $u_1, u_2, u_3$ is given by

$$\mathcal{R}_{12}(u_1, u_2) \mathcal{R}_{13}(u_1, u_3) \mathcal{R}_{23}(u_2, u_3) = \mathcal{R}_{23}(u_2, u_3) \mathcal{R}_{13}(u_1, u_3) \mathcal{R}_{12}(u_1, u_2), \qquad (1)$$

---

[1]We would be very happy to get any info on any other classification works on YBE, in case we missed something.

which resides in $\mathcal{A} \otimes \mathcal{A} \otimes \mathcal{A}$, where $\mathcal{A}$ is the auxiliary vector space. It is an over-determined system for the R-matrix $\mathcal{R}(u, v)$ residing in $\mathcal{A} \otimes \mathcal{A}$, where by using the homomorphisms

$$
\begin{aligned}
\phi_{ij} &: \mathcal{A} \otimes \mathcal{A} \to \mathcal{A} \otimes \mathcal{A} \otimes \mathcal{A} \\
\phi_{12}(x \otimes y) &= a \otimes b \otimes 1 \\
\phi_{23}(x \otimes y) &= 1 \otimes a \otimes b \\
\phi_{13}(x \otimes y) &= a \otimes 1 \otimes b,
\end{aligned}
\tag{2}
$$

we define $\mathcal{R}_{ij}(u_i, u_j) = \phi_{ij}(\mathcal{R}(u_i, u_j))$. For a $N \times N$-dimensional $\mathcal{R}$, one has at most $N^2$ unknowns and a system of at most $N^3$ equations from Eq. (1).

In this work, we consider two spectral parameters by keeping $u_3 = 0$ and impose $\mathcal{R}_{ij}(u_i, u_j) = \mathcal{R}_{ij}(u_i - u_j)$. The YBE becomes

$$
\mathcal{R}_{12}(u)\mathcal{R}_{13}(u + v)\mathcal{R}_{23}(v) = \mathcal{R}_{23}(v)\mathcal{R}_{13}(u + v)\mathcal{R}_{12}(u)
\tag{3}
$$

after replacing $u_1 \to u + v$ and $u_2 \to v$ for notation. It is also known as the different form of the YBE. We also define Eq. (3) in the set of relations as follows:

$$
\mathcal{R}_{\text{YBE}}(u, v) = \mathcal{R}_{12}(u)\mathcal{R}_{13}(u + v)\mathcal{R}_{23}(v) - \mathcal{R}_{23}(v)\mathcal{R}_{13}(u + v)\mathcal{R}_{12}(u),
\tag{4}
$$

where $\mathcal{R}_{\text{YBE}}(u, v) = \mathbf{0}$. At the lowest dimension, we have $\mathcal{A} \simeq \mathbb{C}^2$. The R-matrix then becomes a $4 \times 4$ matrix with at most 16 unknowns to be solved for a system of at most 64 equations.

## 2.2  Symmetries

The symmetries of the R-matrix are the transformations that keep the YBE invariant. We follow the notation in Ref. [21] to identify the relevant symmetries of a $N^2 \times N^2$ R-matrix, where $N$ is the dimension of the auxiliary space

$$
\mathcal{R}(u_a, u_b) = \sum \mathcal{R}_{ij}^{kl}(u_a, u_b)E_{jl} \otimes E_{ik}, \ \ E_{ij} = [(\delta_{ai}\delta_{bj})], \ a, b \in \{1, 2, \dots, N^2\},
\tag{5}
$$

which we call the Hietarinta notation of the R-matrix. Then Eq. (1) is written in the following index notation:

$$
\mathcal{R}_{j_1 j_2}^{k_1 k_2}(u_1, u_2)\mathcal{R}_{k_1 j_3}^{l_1 k_3}(u_1, u_3)\mathcal{R}_{k_2 k_3}^{l_2 l_3}(u_2, u_3) = \mathcal{R}_{j_2 j_3}^{k_2 k_3}(u_2, u_3)\mathcal{R}_{j_1 k_3}^{k_1 l_3}(u_1, u_3)\mathcal{R}_{k_1 k_2}^{l_1 l_2}(u_1, u_2),
\tag{6}
$$

where repeated indices mean summation. The following index notation of the equation reveals the essential symmetries on the R-matrix, which are

1. $\mathcal{R}_{ij}^{kl} \to \mathcal{R}_{kl}^{ij}$                        [Transposition]

2. $\mathcal{R}_{ij}^{kl} \to \mathcal{R}_{i+n \bmod N, \, j+n \bmod N}^{k+n \bmod N, \, l+n \bmod N}$      [Index incremention]

3. $\mathcal{R}_{ij}^{kl} \to \mathcal{R}_{ji}^{lk}$                        [Inversions]

along with the similarity transformation of the R-matrix with the multiplicity freedom

$$
\mathcal{R} \to g(K \otimes K)\mathcal{R}(K \otimes K)^{-1};
\tag{7}
$$

for some non-singular $N \times N$ matrix $K$ and general function $g$. These symmetries enable us to reduce the number of solutions while computing the R-matrices solving YBE. For brevity, we refer to transposition as $P$-transform, index incrementing as $C$-transform, and inversion as $T$-transform.

## 2.3 Differential and single-variable algebraic YB relations

The YBE is a set of cubic functional equations. One of the possible methods used for the solution of functional equations is to differentiate functional equations and to solve obtained differential equations [23]. However, in our approach, we solve the differential YBE and one-variable algebraic equations simultaneously. Let us define $D[M(x,y),x]$ as the total derivative of the Matrix function $M(x,y)$ on $x$. Then,

$$D[\mathcal{R}_{\text{YBE}}(u,v),u], \quad D[\mathcal{R}_{\text{YBE}}(u,v),v] \tag{8}$$

are the examples of differential YB relations, which are always equated with the zero matrix **0**. Then we consider the following list of relations:

$$\begin{aligned}
&\mathcal{R}_{YBE}(0,u), \quad \mathcal{R}_{YBE}(u,0), \\
&D[\mathcal{R}_{YBE}(u,0),u], \quad D[\mathcal{R}_{YBE}(0,u),u], \\
&D[\mathcal{R}_{YBE}(u,v),v]_{v\to 0}, \quad D[\mathcal{R}_{YBE}(v,u),v]_{v\to 0},
\end{aligned} \tag{9}$$

where we identify $\mathcal{R}(0,0) \equiv \mathcal{R}(0)$ as the initial condition for solving the differential equations from the above relation set. To obtain the first set of differential relation, we differentiate Eq. (3) with respect to $v$ and setting $v$ to 0

$$\begin{aligned}
D[\mathcal{R}_{YBE}(u,v),v]_{v\to 0} = &\mathcal{R}_{12}(u)\,\partial_v\mathcal{R}_{13}(u+v)_{v\to 0}\,\mathcal{R}_{23}(0) + \mathcal{R}_{12}(u)\,\mathcal{R}_{13}(u)\,\partial_v\mathcal{R}_{23}(v)_{v\to 0} \\
&- \mathcal{R}_{23}(0)\,\mathcal{R}_{13}(u+v)_{v\to 0}(u)\,\mathcal{R}_{12}(u) - \partial_v\mathcal{R}_{23}(v)_{v\to 0}(0)\,\mathcal{R}_{13}(u)\,\mathcal{R}_{12}(u),
\end{aligned} \tag{10}$$

and similarly by permuting $u$ with $v$ before the differentiation

$$\begin{aligned}
D[\mathcal{R}_{YBE}(v,u),v]_{v\to 0} = &\mathcal{R}_{12}(0)\,\mathcal{R}_{13}(u+v)_{v\to 0}\,\mathcal{R}_{23}(u) + \partial_v\mathcal{R}_{12}(v)_{v\to 0}\,\mathcal{R}_{13}(u)\,\mathcal{R}_{23}(u) \\
&- \mathcal{R}_{23}(u)\,\mathcal{R}_{13}(u+v)_{v\to 0}\,\mathcal{R}_{12}(0) - \mathcal{R}_{23}(u)\,\mathcal{R}_{13}(u)\,\partial_v\mathcal{R}_{12}(v)_{v\to 0}.
\end{aligned} \tag{11}$$

These relations from Eqs. (10) and (11) only contains the spectral parameter $u$. The set of algebraic one-variable relations is obtained from Eq. (4) by nulling any of the spectral parameters

$$\mathcal{R}_{YBE}(u,0) = \mathcal{R}_{12}(u)\mathcal{R}_{13}(u)\mathcal{R}_{23}(0) - \mathcal{R}_{23}(0)\mathcal{R}_{13}(u)\mathcal{R}_{12}(u), \tag{12}$$

$$\mathcal{R}_{YBE}(0,u) = \mathcal{R}_{12}(0)\mathcal{R}_{13}(u)\mathcal{R}_{23}(u) - \mathcal{R}_{23}(u)\mathcal{R}_{13}(u)\mathcal{R}_{12}(0). \tag{13}$$

Finally, we obtain the second set of differential relations from Eqs. (12) and (13) by differentiating with respect to $u$

$$\begin{aligned}
D[\mathcal{R}_{YBE}(u,0),u] = &\mathcal{R}'_{12}(u)\mathcal{R}_{13}(u)\mathcal{R}_{23}(0) + \mathcal{R}_{12}(u)\mathcal{R}'_{13}(u)\mathcal{R}_{23}(0) \\
&- \mathcal{R}_{23}(0)\mathcal{R}'_{13}(u)\mathcal{R}_{12}(u) - \mathcal{R}_{23}(0)\mathcal{R}_{13}(u)\mathcal{R}'_{12}(u),
\end{aligned} \tag{14}$$

$$\begin{aligned}
D[\mathcal{R}_{YBE}(0,u),u] = &\mathcal{R}_{12}(0)\mathcal{R}'_{13}(u)\mathcal{R}_{23}(u) + \mathcal{R}_{12}(0)\mathcal{R}_{13}(u)\mathcal{R}'_{23}(u) \\
&- \mathcal{R}'_{23}(u)\mathcal{R}_{13}(u)\mathcal{R}_{12}(0) - \mathcal{R}_{23}(u)\mathcal{R}'_{13}(u)\mathcal{R}_{12}(0).
\end{aligned} \tag{15}$$

## 2.4 Initial YBE relation

Both sets of differential YBEs (10)–(11) and (14)–(15) contain $R_{ij}(0)$ which also satisfies the constant YBE

$$\mathcal{R}_{12}(0)\mathcal{R}_{13}(0)\mathcal{R}_{23}(0) = \mathcal{R}_{23}(0)\mathcal{R}_{13}(0)\mathcal{R}_{12}(0) \tag{16}$$

The differential YBEs also contain $R'_{ij}(0)$ as parameters. To solve for them we use Eqs. (14)–(15) and set the spectral parameter $u$ towards zero, leading to the below relations

$$D[\mathcal{R}_{YBE}(u,0),u]_{u\to 0} = \mathcal{R}'_{12}(u)_{u\to 0}\mathcal{R}_{13}(0)\mathcal{R}_{23}(0) + \mathcal{R}_{12}(0)\mathcal{R}'_{13}(u)_{u\to 0}\mathcal{R}_{23}(0)$$
$$- \mathcal{R}_{23}(0)\mathcal{R}'_{13}(u)_{u\to 0}\mathcal{R}_{12}(0) - \mathcal{R}_{23}(0)\mathcal{R}_{13}(0)\mathcal{R}'_{12}(u)_{u\to 0}, \qquad (17)$$
$$D[\mathcal{R}_{YBE}(0,u),u]_{u\to 0} = \mathcal{R}_{12}(0)\mathcal{R}'_{13}(u)_{u\to 0}\mathcal{R}_{23}(0) + \mathcal{R}_{12}(0)\mathcal{R}_{13}(0)\mathcal{R}'_{23}(u)_{u\to 0}$$
$$- \mathcal{R}'_{23}(u)_{u\to 0}\mathcal{R}_{13}(0)\mathcal{R}_{12}(0) - \mathcal{R}_{23}(0)\mathcal{R}'_{13}(u)_{u\to 0}\mathcal{R}_{12}(0) \qquad (18)$$

We refer to Eqs. (17)–(18) as initial YBE relations. Setting $u = 0$ in Eqs. (10)–(11) also give the same relations.

## 2.5 Constant solutions of lowest dimension

From Ref. [21], we gather all enlisted invertible constant solutions of the YBE of the lowest dimensions, which we refer them into classes with a label prefixed by "H". Each class denote to a initial condition for solving the YB relations in section 2.3 and 2.4. For brevity, we replace zeros in the matrices with a dot.

Table 1: Invertible and constant solutions of the YBE

| Class | R-matrix |
|-------|----------|
| H31 | $\begin{bmatrix} 1 & \cdot & \cdot & \cdot \\ \cdot & q & \cdot & \cdot \\ \cdot & \cdot & p & \cdot \\ \cdot & \cdot & \cdot & s \end{bmatrix}$ |
| H23 | $\begin{bmatrix} 1 & q & p & s \\ \cdot & 1 & \cdot & p \\ \cdot & \cdot & 1 & q \\ \cdot & \cdot & \cdot & 1 \end{bmatrix}$ |
| H21, H22 = H2x | $\begin{bmatrix} 1 & \cdot & \cdot & \cdot \\ \cdot & q & \cdot & \cdot \\ \cdot & 1-qp & p & \cdot \\ \cdot & \cdot & \cdot & k \end{bmatrix}$ |
| H14 | $\begin{bmatrix} \cdot & \cdot & \cdot & q \\ \cdot & \cdot & 1 & \cdot \\ \cdot & 1 & \cdot & \cdot \\ p & \cdot & \cdot & \cdot \end{bmatrix}$ |
| H13 | $\begin{bmatrix} 1 & -p & p & pq \\ \cdot & 1 & \cdot & -q \\ \cdot & \cdot & 1 & q \\ \cdot & \cdot & \cdot & 1 \end{bmatrix}$ |
| H12 | $\begin{bmatrix} 1 & \cdot & \cdot & k \\ \cdot & q & \cdot & \cdot \\ \cdot & 1-q & 1 & \cdot \\ \cdot & \cdot & \cdot & -q \end{bmatrix}$ |

H11
$$\begin{bmatrix} 1+2q-q^2 & \cdot & \cdot & 1-q^2 \\ \cdot & 1+q^2 & 1-q^2 & \cdot \\ \cdot & 1-q^2 & 1+q^2 & \cdot \\ 1-q^2 & \cdot & \cdot & 1-2q-q^2 \end{bmatrix}$$

H02
$$\begin{bmatrix} 1 & \cdot & \cdot & 1 \\ \cdot & -1 & 1 & \cdot \\ \cdot & 1 & 1 & \cdot \\ -1 & \cdot & \cdot & 1 \end{bmatrix}$$

H01
$$\begin{bmatrix} 1 & \cdot & \cdot & 1 \\ \cdot & -1 & \cdot & \cdot \\ \cdot & \cdot & -1 & \cdot \\ \cdot & \cdot & \cdot & 1 \end{bmatrix}$$

Similarly, we also gather all the rank-3 constant solutions below

Table 2: Rank-3 constant solutions of the YBE

| Class | R-matrix |
|-------|----------|
| H15 | $\begin{bmatrix} p+q & \cdot & \cdot & \cdot \\ \cdot & q & \cdot & q \\ \cdot & \cdot & p+q & \cdot \\ \cdot & p & \cdot & p \end{bmatrix}$ |
| H16 | $\begin{bmatrix} \cdot & p & p & \cdot \\ \cdot & \cdot & k & q \\ \cdot & k & \cdot & q \\ \cdot & \cdot & \cdot & \cdot \end{bmatrix}$ |
| H04 | $\begin{bmatrix} 1 & \cdot & \cdot & \cdot \\ \cdot & \cdot & \cdot & 1 \\ \cdot & 1 & \cdot & \cdot \\ \cdot & \cdot & \cdot & 1 \end{bmatrix}$ |
| H05 | $\begin{bmatrix} 1 & \cdot & \cdot & \cdot \\ \cdot & \cdot & 1 & \cdot \\ \cdot & 1 & \cdot & \cdot \\ \cdot & \cdot & \cdot & \cdot \end{bmatrix}$ |

## 3  Results

### 3.1  Notation

We present the final set of R-matrices for Eq. (3) that are obtained from our analysis. First we begin with providing full-rank solutions where we identify 2 constant R-Matrices which do not allow for further Baxterization. We further identify 10 non-constant cases, including the trivial diagonal case. Initially our algorithm provide many more solutions which we list in the Appendix B. By using symmetries in 2.2, we further distillate them into the final list.

Furthermore, we also provide rank-3 solution that our algorithm has managed to obtain. There are many lower-rank solutions which are yet to be reported.

We use $c_1, c_2, \ldots$ etc. as the notation for general spectral-independent coefficients alongside with $p_0, q_0$ and $k_0$ from the initial constraints. We also employ $r_1(u), r_2(u), \ldots$ for spectral-dependent terms. These results are exhaustive and can be further searched from our algorithm.

## 3.2 Identification of equivalent cases

**Similarity transformation**

Below we consider different solutions to be equivalent, if they transform into each other using $P$-, $C$- and $T$-transformations defined in section 2.2, as well as similarity transformation (7). The $K$-matrix belongs to $SL(2, \mathbb{C})$-group and can be parametrised as

$$K = K(a, b, c) = \begin{pmatrix} a & 0 \\ 0 & 1/a \end{pmatrix} \begin{pmatrix} 1 & 0 \\ c & 1 \end{pmatrix} \begin{pmatrix} 1 & b \\ 0 & 1 \end{pmatrix}. \tag{19}$$

A list of $a, b, c$ parameters for every case mentioned below can be provided.

**Inversion transformation**

By taking the inversion on both sides of Eq. (1) we find

$$\mathcal{R}_{23}^{-1}(u_2, u_3) \mathcal{R}_{13}^{-1}(u_1, u_3) \mathcal{R}_{12}^{-1}(u_1, u_2) = \mathcal{R}_{12}^{-1}(u_1, u_2) \mathcal{R}_{13}^{-1}(u_1, u_3) \mathcal{R}_{23}^{-1}(u_2, u_3), \tag{20}$$

which is the YBE (1) for the inverse matrix $\mathcal{R}^{-1}$. By doing this, the R-matrix $\mathcal{R}$ and its inverse $\mathcal{R}^{-1}$ (irrespective of redefinition of parameters and terms) are considered equivalent in our classification.

## 3.3 Full rank solution

### 3.3.1 Constant $R$-matrices.

1.

$$\begin{pmatrix} 1 & 0 & 0 & \frac{q_0^2-1}{q_0^2-2q_0-1} \\ 0 & \frac{q_0^2+1}{-q_0^2+2q_0+1} & \frac{q_0^2-1}{q_0^2-2q_0-1} & 0 \\ 0 & \frac{q_0^2-1}{q_0^2-2q_0-1} & \frac{q_0^2+1}{-q_0^2+2q_0+1} & 0 \\ \frac{q_0^2-1}{q_0^2-2q_0-1} & 0 & 0 & \frac{q_0^2+2q_0-1}{q_0^2-2q_0-1} \end{pmatrix}$$

2.

$$\begin{pmatrix} 1 & 0 & 0 & k_0 \\ 0 & 1 & 1-p_0 & 0 \\ 0 & 0 & p_0 & 0 \\ 0 & 0 & 0 & -p_0 \end{pmatrix}$$

### 3.3.2 New non-constant $R$-matrices.

1.

$$\begin{pmatrix} 1 & 0 & 0 & r_1(u) \\ 0 & -1 & 0 & 0 \\ 0 & 0 & -1 & 0 \\ 0 & 0 & 0 & 1 \end{pmatrix}$$

2.

$$\begin{pmatrix} 1 & 0 & 0 & e^{c_1 u} \\ 0 & -1 & e^{c_1 u} & 0 \\ 0 & e^{c_1 u} & 1 & 0 \\ -e^{c_1 u} & 0 & 0 & 1 \end{pmatrix}$$

3.

$$\begin{pmatrix} 1 & 0 & 0 & r_1(u) \\ 0 & 1 & 0 & 0 \\ 0 & 0 & 1 & 0 \\ 0 & 0 & 0 & -1 \end{pmatrix}$$

4.

$$\begin{pmatrix} 1 & 0 & 0 & k_0 e^{c_1 u} \\ 0 & -1 & 2e^{c_1 u} & 0 \\ 0 & 0 & 1 & 0 \\ 0 & 0 & 0 & 1 \end{pmatrix}$$

5.

$$\begin{pmatrix} 1 & r_1(u) & r_2(u) & r_3(u) \\ 0 & 1 & 0 & r_2(u) \\ 0 & 0 & 1 & r_1(u) \\ 0 & 0 & 0 & 1 \end{pmatrix}$$

6.

$$\begin{pmatrix} 1 & -p_0 & p_0 & c_1 u + p_0 q_0 \\ 0 & 1 & 0 & -q_0 \\ 0 & 0 & 1 & q_0 \\ 0 & 0 & 0 & 1 \end{pmatrix}$$

7.

$$\begin{pmatrix} 0 & 0 & 0 & p_0 \\ 0 & 0 & r_1(u) & 0 \\ 0 & r_1(u) & 0 & 0 \\ 1 & 0 & 0 & 0 \end{pmatrix}$$

8.

$$\begin{pmatrix} 1 & 0 & 0 & 0 \\ 0 & p_0 & e^{c_1 u}(1-p_0 q_0) & 0 \\ 0 & 0 & q_0 & 0 \\ 0 & 0 & 0 & -p_0 q_0 \end{pmatrix}$$

9.

$$\begin{pmatrix} 1 & 0 & 0 & 0 \\ 0 & p_0 & e^{c_1 u}(1-p_0 q_0) & 0 \\ 0 & 0 & q_0 & 0 \\ 0 & 0 & 0 & 1 \end{pmatrix}$$

10. Trivial case.

$$\begin{pmatrix} 1 & 0 & 0 & 0 \\ 0 & r_1(u) & 0 & 0 \\ 0 & 0 & r_2(u) & 0 \\ 0 & 0 & 0 & r_3(u) \end{pmatrix}$$

## 3.4 Rank 3 solution

### 3.4.1 Constant rank 3 $R$-matrices

1.

$$\begin{pmatrix} 1 & 0 & 0 & 0 \\ 0 & 1 & 0 & 0 \\ 0 & 0 & \frac{q_0}{p_0+q_0} & \frac{q_0}{p_0+q_0} \\ 0 & 0 & \frac{p_0}{p_0+q_0} & \frac{p_0}{p_0+q_0} \end{pmatrix}$$

### 3.4.2 New non-constant rank 3 $R$-matrices

1.

$$\begin{pmatrix} 1 & 0 & 0 & 0 \\ 0 & 0 & e^{c_1 u} & 1-e^{c_1 u} \\ 0 & 0 & 0 & 1 \\ 0 & 0 & 0 & 1 \end{pmatrix}$$

2.

$$\begin{pmatrix} 0 & 0 & 0 & 0 \\ 0 & 0 & e^{c_2 u} & 0 \\ 0 & e^{c_1 u} & 0 & 0 \\ 0 & 0 & 0 & 1 \end{pmatrix}$$

3.

$$\begin{pmatrix} 0 & 1 & e^{c_1 u} & r_1(u) \\ 0 & 0 & k_0 e^{c_1 u} & q_0 e^{c_1 u} \\ 0 & k_0 & 0 & q_0 \\ 0 & 0 & 0 & 0 \end{pmatrix}$$

# 4 Algorithm

In this section, we provide the details of the involved workflow and showcase the main aspects of our algorithm. We have used the `Mathematica` software to build the code base.

First we enlist the important stages of the algorithm. From Table 1, we set the initial condition $\mathcal{R}(0)$ and prepare the relations (Stage 1). Then the algorithm solves the initial YB relation in Sec. 2.4 (Stage 2). The one-variable algebraic and differential YB relation are handled (Stage 3). Solutions from both steps are then substituted to Eq. (3) and any residual equation are resolved if they exist (Stage 4).

The procedure to resolve initial YB relations and the differential ones are similar. Hence we address the workflow for the resolution of the relations (9), while presenting the description of Stage 2 in Appendix.

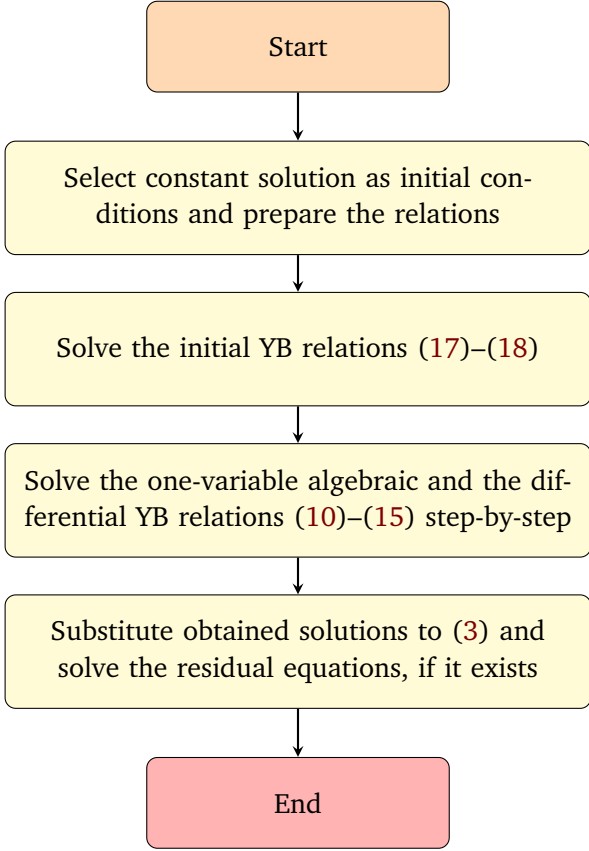

Figure 1: Sequence order of different stages in the algorithm

## 4.1 Stage 1 : Preparation of relations

In our work, the initial conditions $\mathcal{R}(0)$ chosen for resolving Eq. (9) from Table 1 are considered invertible which enforce the conditions on the free parameters of the constant R-matrices. It is not necessary for the algorithm as we also chose non-invertible cases from Table 2 to obtain various rank-3 solutions too. A complete set of relations are prepared and separated in the following manner:

1. Algebraic relations $\mathcal{A}(u) : \mathcal{R}_{YBE}(0, u), \mathcal{R}_{YBE}(u, 0)$

2. Differential relations $\mathcal{D}(u) : D[\mathcal{R}_{YBE}(u, 0), u], D[\mathcal{R}_{YBE}(0, u), u], D[\mathcal{R}_{YBE}(u, v), v]_{v \to 0},$
   $D[\mathcal{R}_{YBE}(v, u), v]_{v \to 0}$

3. Initial condition of differential relations $\mathcal{D}(0)$

Relations in $\mathcal{D}(u)$ and $\mathcal{D}(0)$, which has no differential form is shifted in $\mathcal{A}(u)$. After then $\mathcal{D}(0)$ is resolved in Stage 2 of the algorithm, effectively solving Eqs.(17)–(18). The majority of the algorithm deals with resolving $\mathcal{A}(u)$ and $\mathcal{D}(u)$ which we describe the following sections.

## 4.2 Stage 3 : Solving one-variable algebraic and the differential YB relations

The main idea behind our proposal is to solve YB relations step-by-step in a manner that we avoid obtaining repeated solutions, which we encounter while resolving an over-determined system of equations. For every step, we choose *few* and *simple* equations from algebraic and differential relations and solve them. Since this leads to multiple solution in each step, we get result branches which can be visualised as in Fig. 2.

There are two different means of resolution for the relations: First to solve the differential ones and secondly to consider them as system of polynomials that are reducible through Gröbner basis calculations. Solving them exclusively through one method is not feasible as it leads to excessive computational time. Furthermore there is no systematic means to control the complexity of the obtained solutions even after solving them, which adds to difficulties in classifying them further.

Hence to have a substantial control on the resolution of the relations, we choose a trade-off between these two method by considering a branching algorithm which searches for *solution vertices* in a graph of various *search branches*. The algorithm begins with the initiation vertex which carries all the relations of Eq. (9) alongside with the solutions of the initial YB relations. A subset of relations are solved to generate subcases. Each of them then represent *search branches* which are recursively simplified to obtain a valid solution of the YBE until exhaustion.

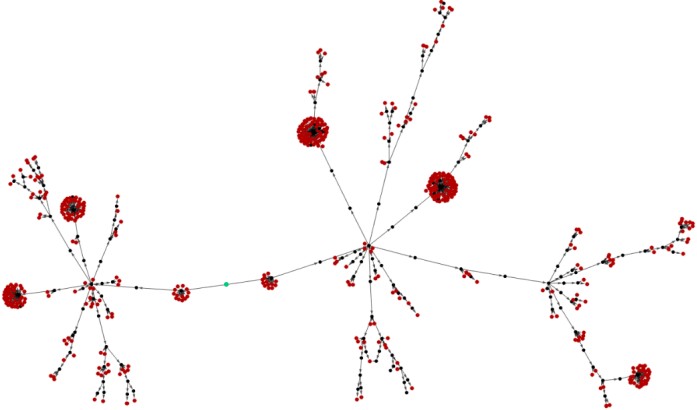

Figure 2: A graph of search branches produced by the algorithm while searching for solutions of the YBE for an initial constraint (H14). The red dots indicate terminated vertices which may indicate a valid solution or an invalid (often indeterminate) one. The green dot is the *initiation vertex* from where the algorithm initiated.

**Initialisation**

The algorithm begins with the initialization vertex which carries all relations $\mathcal{A}(u)$ and $\mathcal{D}(u)$. It decides to evaluate a subset of relations from the algebraic relations $\mathcal{A}(u)$ in given accordance

- If the relations in $\mathcal{A}(u)$ is less than a specified range $N_{\text{lim}}$, then we choose to compute their Gröbner basis first.

- If it is not the case, then a subset of relations having a linear or factorised form is considered. It is under always under a limit of its *complexity length* which is determined through the number of the terms in their polynomial expansion. The maximal length is denoted by $N_{\text{term}}$.

The chosen set is further solved by employing `Reduce/Solve` method in Mathematica. The results go through a series of evaluations for checking if they solve the complete relation set. Those which do so are labelled as solutions of Eq. (3). Result vertices with indeterminate terms or invalid logic are terminated by assigning them as `Stopped`. The rest of the search branches are considered for the next iteration. To solve the issue of repeated solutions, any branches having the identical contents in their relation set are then merged.

**Iteration**

The collection of unresolved branches go through a similar process of simplification as described in the former section. However there are central differences from the initialisation stage. At every iteration step the algebraic $\mathcal{A}(u)$ and differential $\mathcal{D}(u)$ relations are separated similarly as in the previous step as the algorithm continues the loop. To simplify every branch for further subcases, the algorithm considers the following scheme:

- A subset of relations from $\mathcal{A}(u)$ having a linear or factorised form and satisfying their complexity length lesser than $N_{\text{term}}$ is evaluated.

- If no such relations are available then the algorithm tries computing Gröbner basis for the relations in $\mathcal{A}(u)$ if it is lesser than or equal to $N_{\text{lim}}$.

- If the relations in $\mathcal{A}(u)$ is larger than $N_{\text{lim}}$, then the algorithm includes the shortest differential relation from $\mathcal{D}(u)$ whose complexity length is lesser than $N_{\text{diff}}$ and uses `DSolve[]` method.

- Otherwise it proceeds for the next unresolved branch.

The algorithm goes through the process of labelling resolved solutions as *temporarily stopped* (TempStopped) for further finalisation, terminating indeterminate/invalid cases, merging identical results and building new search branches. It reiterates until there are no new unresolved branches possible. In the end, the computations are systematically complied in an `Association` graph object containing all details of the evaluation. To summarise the iteration, we refer to Fig. 3.

**Finalisation**

All completed solution branches are collated and verified of being a solution of (3). Any residual case where it does not resolve the YBE is solved in Stage 4. The ones which solves them are then labelled as (Finalized). After then by employing the symmetries of the R-Matrices, any redundant cases are removed.

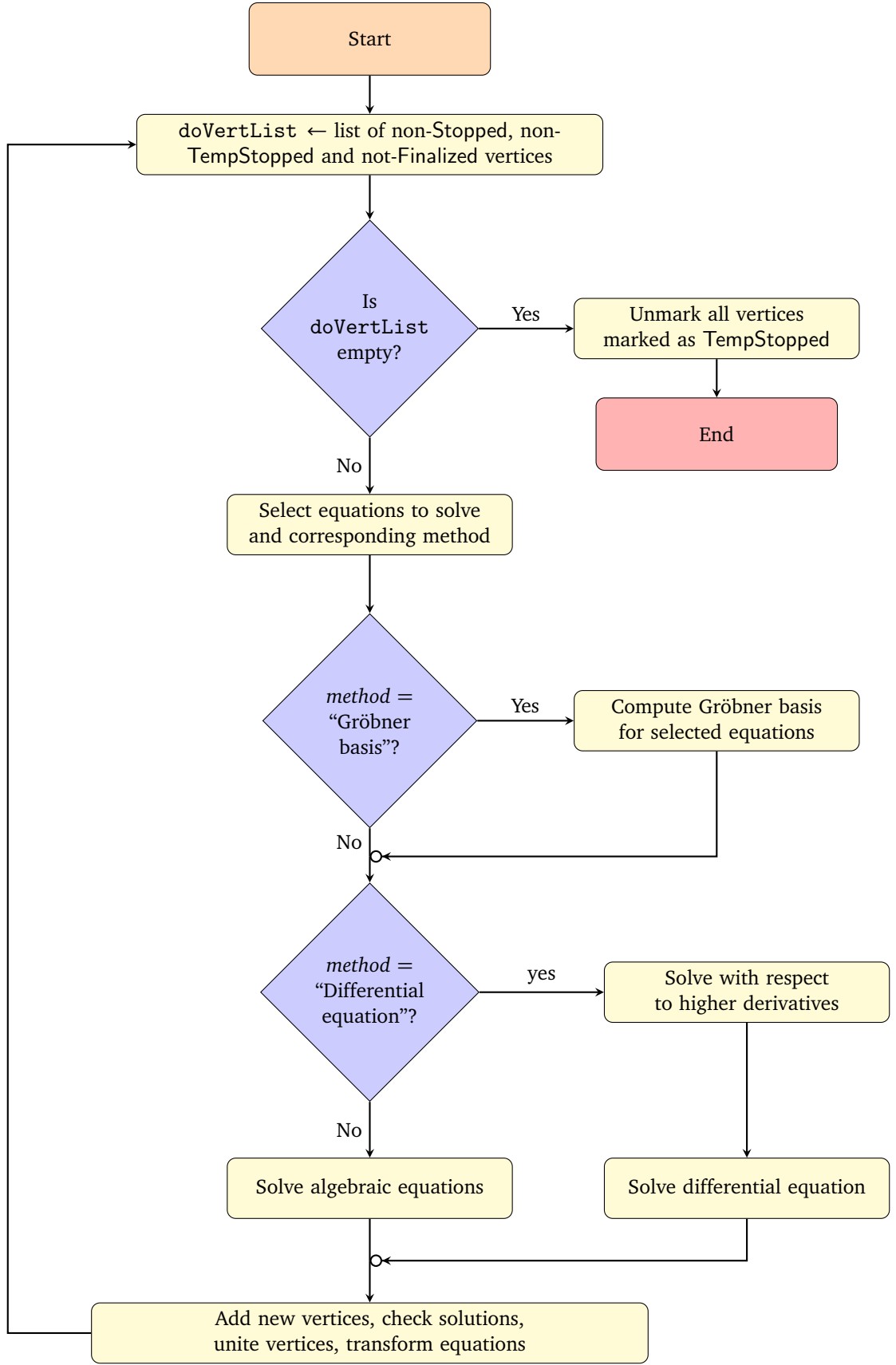

Figure 3: Algorithm flowchart for the stage 3.

### 4.3 Algorithm's determinants and its limitations

We summarise all the constraints that are involved in the execution of the algorithm with their default parameters used

1. The complexity length of a algebraic relation, $N_{\text{term}}$ (4)

2. The complexity length of a differential relation, $N_{\text{diff}}$ (8)

3. The number of relations for Gröbner basis computations, $N_{\text{lim}}$ (24)

4. Time limit for running Gröbner basis computations, $t_{\text{gb}}$ (20 mins)

5. Time limit for running `Reduce[]` method, $t_{\text{rd}}$ (5 min)

Alongside with the initial condition $\mathcal{R}(0)$, we have the complete set of determinants of the algorithm. One may adjust these parameters for an extensive algorithmic search, which further depends on the available hardware configurations on which the programs are executed.

**Advantages**

We then infer the limitations of the algorithm from these determinants. Using $N_{\text{term}}$ and $N_{\text{diff}}$ in a larger setting will not influence the algorithm significantly, as it leads to lesser iterative steps for algebraic simplification at the cost of increasing the number of search branches per execution.

The important constraints are $N_{\text{lim}}$ and both time limits. Since they cap the scope of solving difficult time-consuming relations, the algorithm may not be able to finish resolving certain terminated branches. An notable feature of this process is that all terminated search branches from the final output of the program can be re-executed with broader determinants. Hence one can extend the incomplete tedious algorithmic search in the future.

**Shortcomings and Scopes**

The shortcomings comes mainly from developing custom pattern-based filters to identify complicated set of relations which are either used to avoid untenable search branches or to simplify them further. These methods are additionally developed which then provides future scopes for customising the algorithm.

In the context of this work, we have avoided dealing with nonlinear relations (those with power terms) explicitly and considered simplifying them through Gröbner computations. Square-root sub-expressions are tended by assuming them to be positive, as every branch cut of such terms are already considered in the `Reduce[]` method. In this manner, the algorithm only dealt with linearised relations from where the R-matrices are computed. Hence we could present a *sufficiently exhaustive* search for all spectral solutions of the YBE.

## 5 Conclusion and Outlooks

In this paper, we have developed an algorithm that allows developing of a broader classification scheme for solutions of the celebrated YBE than what has been used before. We have demonstrated this method on an example of $4 \times 4$ matrix solutions of YBE, which corresponds to the two-qubit case. We have found a number of new solutions, which after applications of symmetry-based reduction leads a much fewer classes.

We note some further directions here. The first is the application to higher-dimensional matrix solutions of YBE. The second direction is the application to non-matrix solutions of YBE.

Third, applications of obtained R-matrices to the "quasiclassical" limit (known as r-matrices) satisfying "classical" YBE. Finally, it is interesting to consider physical applications in terms of spin chains (possibly non-Hermitian; e.g., see Ref. [33]) and generalized (also, possibly non-Hermitian) quantum circuits.

## Acknowledgements

**Funding information**   Work of SB and VG is partially supported by the Delta Institute for Theoretical Physics (DITP). DITP consortium, a program of the Netherlands Organization for Scientific Research (NWO) is funded by the Dutch Ministry of Education, Culture and Science (OCW). A.K.F. is supported by the Priority 2030 program at the National University of Science and Technology "MISIS" under the project K1-2022-027 and the Russian Roadmap on Quantum Computing (Contract No. 868-1.3-15/15-2021).

## A   Pseudocodes for solving one variable differential and algebraic relations

### A.1   Data structures

dRRRList[$i$] — a list of differential YBEs at step $i$.
algRRRList[$i$] — a list of one-variable algebraic YBEs at step $i$.
initDRRRList[i] — a list of initial YBEs at step $i$. RmGraph — a graph of solution process. Every vertex of the graph RmGraph has boolean properties Stopped, TempStopped and Finalized or not, that means we have a contradiction or (partially) final solution correspondingly.

*Temporal variables:*
tmpEqnsList[$i$] — a list of equations to be solved for vertex $i$.
doVertList — a list of vertices of the graph RmGraph to be considered during the current cycle (see below ...).
tmpEqnsList[$i$] — a list of equations to be solved for vertex $i$.
tmpEqnsList[$i$] — a list of equations to be solved for vertex $i$.
*doVertList* — a list of vertices of the graph *RmGraph* to be considered during the current cycle
*method* — global variable.

### A.2   Algorithm of the workflow

**Algorithm 1** Stage 3

**Description**: Algorithm of Stage 3

   Local doNewVertList                                            ▷ *To store new vertices*
1▷  doVertList ← GETTODOVERTICES(RmGarph)
2▷  **while** doVertList is not empty **do**                                          ▷ *x*
3▷    *method* ← SELECTEQNSTOSOLVE(dRRRList, algRRRList, tmpEqnsList,
     doVertList)                    ▷ *Select method to solve, select equations to solve*
4▷    REDUCEVERTEXEQNS(tmpEqnsList, doVertList, rSolution)
5▷    PROCSOLUTIONS(RmGarph, rSolution, algRRRList, dRRRList, doVertLst)
6▷    doNewVertList ← GETNEWVERTICES(RmGarph, doVertList)

7▸       UNITEVERTICES(RmGarph, doNewVertList)

8▸       doNewVertList ← GETNEWVERTICES(RmGarph, doVertList)

9▸       TRANSFORMEQUATIONSLISTS(algRRRList, dRRRList, doNewVertList)

10▸      doVertList ← GETTODOVERTICES(RmGarph)

11▸ **end while**

12▸ **for all** $i \in$ RmGraph and marked as TempStopped **do**

13▸      Unmark vertex $i$ as TempStopped

14▸ **end for**

---

**Algorithm 2** Code of GetToDoVertices function

---

**Description**:

**Input**: RmGarph

**Output**: list of vertices to be processed

**Require**: RmGarph exists

1▸ **function** GETTODOVERTICES(RmGarph)

2▸      Local tmpList ← empty list

3▸      **for each** $i \in$ vertices of RmGraph **do**

4▸         **if** (vertex $i$ has not child vertices) **and** (vertex $i$ is not marked as Stopped or TempStopped or Finalized) **then**

5▸           tmpList ← tmpList $\cup \{i\}$                ▷ *Append i to* tmpList

6▸         **end if**

7▸      **end for**

8▸      **return** tmpList

9▸ **end function**

---

**Algorithm 4** ReduceVertexEqns

---

**Description**: Solve selected equations

**Input**: tmpEqnsList, dovertList

**Output**: append obtained solutions to rSolution

**Require**: tmpEqnsList[$i$] is not empty $\forall i \in$ doVertList

1▸ **procedure** REDUCEVERTEXEQNS(tmpEqnsList, doVertList, rSolution)

2▸      Local *tmpDEqn*, tmpDEqns

        ▷ *To store current differential equation to be algebraically solved with respect to higher derivatives and all differential equations*

3▸      **for each** $i \in$ doVertList **do**

4▸         **if** *method* ≠ "Differential equation method" **then**

5▸           rSolution[$i$] ← SOLVE(tmpEqnsList[$i$])      ▷ *Solve algebraic equations*

6▸         **else**

7▸           tmpDEqns ← SOLVE(tmpEqnsList[$i$])

            ▷ *Algebraically solve differential equation with respect to higher derivatives*

8▸           **for each** *tmpDEqn* in tmpDEqns **do**

9▸             rSolution[$i$] ← rSolution[$i$] $\cup$ DIFFSOLVE(*tmpDEqn*)

            ▷ *Solve differential equation and append obtained solution to* rSolution[$i$]

10▸           **end for**

11▸         **end if**

12▸         **if** rSolution[$i$] is empty list **then**

13▸           Mark vertex $i$ as Stopped                 ▷ *No solutions*

---

**Algorithm 3** Selection of equations to solve and method for solving

---

**Description**: Select equations to solve and corresponding method
**Input**: dRRRList, algRRRList, doVertList
**Output**: updates tmpEqnsList and set global variable *method*
**Require**: $\forall i \in$ doVertLst $:$ (LENGTH(algRRRList[$i$]) $> 0$) **or** (LENGTH(dRRRList[$i$]) $> 0$)

---

1▶ **procedure** SELECTEQNSTOSOLVE(dRRRList, algRRRList, tmpEqnsList, doVertList)
2▶     Local eqnsCountList          ▷ *contains number of selected equations*
3▶     eqnsCountList ← empty list
4▶     **for each** $i \in$ doVertList **do**
5▶         tmpEqnsList[$i$] ← linear in $\mathcal{R}^{k_1 k_2}_{j_1 j_2}(u)$ equations from algRRRList[$i$] having no more
            than $N_{\text{term}}$ terms in expanded form
6▶         eqnsCountList ← eqnsCountList $\cup$ {LENGTH(tmpEqnsList[$i$])}
                   ▷ *append number of equations in* tmpEqnsList[$i$] *to* eqnsCountList
7▶     **end for**
8▶     **if** eqnsCountList has non-zero elements **then**
9▶         *method* ← "Linear method"          ▷ *Linear method*
10▶     **else**
11▶         **for each** $i \in$ doVertList **do**
12▶             tmpEqnsList[$i$] ← factorizable equations from algRRRList[$i$]
13▶             eqnsCountList ← eqnsCountList $\cup$ {LENGTH(tmpEqnsList[$i$])}
14▶         **end for**
15▶         **if** eqnsCountList has non-zero elements **then**
16▶             *method* ← "Product method"          ▷ *Product method*
17▶         **else**
18▶             **for each** $i \in$ doVertList **do**
19▶                 **if** LENGTH(algRRRList[$i$]) $< N_{\text{lim}}$ **then**
20▶                     tmpEqnsList[$i$] ← algRRRList[$i$]      ▷ $N_{\text{lim}}$ *is a tuning parameter*
21▶                 **else**
22▶                     tmpEqnsList[$i$] ← empty list
23▶                 **end if**
24▶                 eqnsCountList ← eqnsCountList $\cup$ {LENGTH(tmpEqnsList[$i$])}
25▶             **end for**
26▶             **if** eqnsCountList has non-zero elements **then**
27▶                 *method* ← "Gröbner basis method"      ▷ *Gröbner basis method*
28▶             **else**
29▶                 **for each** $i \in$ doVertList **do**
30▶                     tmpEqnsList[$i$] ← one shortest equation from dRRRList[$i$] and having no
                        more than $N_{\text{diff}}$ terms in expanded form
31▶                     eqnsCountList ← eqnsCountList $\cup$ {1}    ▷ *append* 1 *to* eqnsCountList
32▶                 **end for**
33▶                 **if** eqnsCountList has non-zero elements **then**
34▶                     *method* ← "Differential equation method" ▷ *Differential equation method*
35▶                 **else**
36▶                     Break algorithm
37▶                 **end if**
38▶             **end if**
39▶         **end if**
40▶     **end if**
41▶ **end procedure**

---

14▸        **end if**
15▸     **end for**
16▸ **end procedure**

---

**Algorithm 5** Process obtained solutions

**Description**: Add vertices to the graph. One vertex for every new solution.
**Input**: RmGarph, rSolution, algRRRList, dRRRList, doVertLst
**Output**: Updates algRRRList, dRRRList
**Require**: RmGarph exists

1▸ **procedure** PROCSOLUTIONS(RmGarph, rSolution, algRRRList, dRRRList,
                                  doVertLst)
2▸    Local *tmpSol*                          ▷ *Local variable to store current solution*
3▸    $j \leftarrow$ number of vertices of RmGraph
4▸    **for each** $i \in$ doVertList **do**
5▸       **for each** tmpSol $\in$ rSolution[$i$] **do**
6▸          $j \leftarrow j + 1$
7▸          Append new vertex $j$ to graph RmGraph
8▸          Add directed edge from vertex $i$ to child vertex $j$
9▸          algRRRList[$j$] $\leftarrow$ substitute tmpSol to algRRRList[$i$]
10▸         dRRRList[$j$] $\leftarrow$ substitute tmpSol to dRRRList[$i$]
11▸         **if** (obtained solution leads to degenerate or singular $\mathcal{R}$-matrix) **or**
           (algRRRList[$i$] is singular) **or** (algRRRList[$i$] is singular) **then**
12▸            Mark vertex $j$ as Stopped
13▸         **end if**
14▸       **end for**
15▸    **end for**
16▸ **end procedure**

---

**Algorithm 6** Code of GetNewVertices function

**Description**:
**Input**: RmGraph, doVertList
**Output**: list of new vertices
**Require**: RmGarph exists **and** doVertList is not empty

1▸ **function** GETNEWVERTICES(RmGarph, doVertList)
2▸    Local tmpList $\leftarrow$ empty list
3▸    **for each** $i \in$ doVertList **do**
4▸       **for all** $j \in$ child vertices of vertex $i$ of RmGraph **do**
5▸          **if** (vertex $i$ has not child vertices) **and** (vertex $i$ is not marked as Stopped or
           TempStopped or Finalized) **then**
6▸            tmpList $\leftarrow$ tmpList $\cup \{i\}$            ▷ *Append i to* tmpList
7▸          **end if**
8▸       **end for**
9▸    **end for**
10▸    **return** tmpList
11▸ **end function**

---

**Algorithm 7** Code of UniteVertices

**Description**:

**Input**: `RmGraph, doNewVertList`
**Output**: Updates `RmGraph`
**Require**:

1▸ **procedure** UNITEVERTICES(`RmGarph, doNewVertList`)
2▸     Local `tmpList` ← empty list
3▸     **for each** $i \in$ `RmGraph` **do**
4▸         **for all** $j \in$ `doVertList`, $j > i$ **do**
5▸             **if** ($j \notin$ `tmpList`) **and** (edge between vertices $j$ and $i$ does not exist) **then**
6▸                 **if** vertices $i$ and $j$ leads to the same $\mathcal{R}$-matrix **then**
7▸                     `tmpList` ← `tmpList` ∪ $\{j\}$                    ▷ *Append $j$ to* `tmpList`
8▸                     Add directed edge from vertex $j$ to vertex $i$ in `RmGraph`
9▸                 **end if**
10▸             **end if**
11▸         **end for**
12▸     **end for**
13▸ **end procedure**

---

**Algorithm 8** Code of TransformEquationsLists

---

**Description**:
**Input**: `algRRRList, dRRRList, doNewVertList`
**Output**: Updates `algRRRList, dRRRList`
**Require**:

1▸ **procedure** TRANSFORMEQUATIONSLISTS(`algRRRList, dRRRList, doNewVertList`)
2▸     **for each** $i \in$ `doNewVertList` **do**
3▸         Remove constant non-zero factors for every expression in `algRRRList`[$i$]
4▸         Remove zeros from `algRRRList`[$i$]
5▸         Remove constant non-zero factors for every expression in `dRRRList`[$i$]
6▸         Remove zeros from `algRRRList`[$i$]
7▸         **if** some expression in `algRRRList`[$i$] or in `dRRRList`[$i$] is non-zero constant **then**
8▸             Mark vertex $i$ as Stopped
9▸         **end if**
10▸         **if** both `algRRRList` and `dRRRList` are empty lists **then**
11▸             Mark vertex $i$ as TempStopped
12▸         **end if**
13▸     **end for**
14▸ **end procedure**

## A.3 Algorithm and workflow of stage 2

---

**Algorithm 9** Stage 2

---

**Description**: Algorithm of stage 2

    Local doNewVertList                       ▷ *To store new vertices*

1▸  doVertList ← GETTODOVERTICES(RmGarph)

2▸  **while** doVertList is not empty **do**                    ▷ *x*

3▸     *method* ← SELECTINITEQNSTOSOLVE(initDRRRList, tmpEqnsList, doVertList)
           ▷ *Select method to solve, select equations to solve*

4▸     REDUCEVERTEXEQNS(tmpEqnsList, doVertList, rSolution)

5▸     PROCSOLUTIONS(RmGarph, rSolution, algRRRList, dRRRList, initDRRRList,
    doVertLst)

6▸     doNewVertList ← GETNEWVERTICES(RmGarph, doVertList)

7▸     UNITEVERTICES(RmGarph, doNewVertList)

8▸     doNewVertList ← GETNEWVERTICES(RmGarph, doVertList)

9▸     TRANSFORMEQUATIONSLISTS(algRRRList, dRRRList, initDRRRList,
    doNewVertList)

10▸    doVertList ← GETTODOVERTICES(RmGarph)

11▸ **end while**

12▸ **for all** $i \in$ RmGraph and marked as TempStopped **do**

13▸    Unmark vertex $i$ as TempStopped

14▸ **end for**

---

**Algorithm 10** Selection of equations to solve and method for solving at stage 2

---

**Description**: Select equations to solve and corresponding method for soliing $\mathcal{D}(0)$ equations
**Input**: initDRRRList, doVertList
**Output**: updates tmpEqnsList and set global variable *method*
**Require**: $\forall i \in$ doVertLst : LENGTH(initDRRRList[$i$]) > 0

1▸ **procedure** SELECTINITEQNSTOSOLVE(initDRRRList, tmpEqnsList, doVertList)

2▸   Local eqnsCountList               ▷ *contains number of selected equations*

3▸   eqnsCountList ← empty list

4▸   **for each** $i \in$ doVertList **do**

5▸     tmpEqnsList[$i$] ← linear in $\mathcal{R}_{j_1 j_2}^{k_1 k_2}(u)$ equations from initDRRRList[$i$] having no more
     than $N_{\text{term}}$ terms in expanded form

6▸     eqnsCountList ← eqnsCountList ∪ {LENGTH(tmpEqnsList[$i$])}
              ▷ *append number of equations in* tmpEqnsList[$i$] *to* eqnsCountList

7▸   **end for**

8▸   **if** eqnsCountList has non-zero elements **then**

9▸     *method* ← "Linear method"                ▷ *Linear method*

10▸   **else**

11▸     **for each** $i \in$ doVertList **do**

12▸      tmpEqnsList[$i$] ← factorizable equations from initDRRRList[$i$]

13▸      eqnsCountList ← eqnsCountList ∪ {LENGTH(tmpEqnsList[$i$])}

14▸     **end for**

15▸     **if** eqnsCountList has non-zero elements **then**

16▸      *method* ← "Product method"             ▷ *Product method*

17▸     **else**

18▸      **for each** $i \in$ doVertList **do**

19▸       **if** LENGTH(initDRRRList[$i$]) < $N_{\text{lim}}$ **then**

20▸        tmpEqnsList[$i$] ← initDRRRList[$i$]     ▷ $N_{\text{lim}}$ *is a tuning parameter*

```
21▸              else
22▸                  tmpEqnsList[i] ← empty list
23▸              end if
24▸              eqnsCountList ← eqnsCountList ∪ {LENGTH(tmpEqnsList[i])}
25▸          end for
26▸          if eqnsCountList has non-zero elements then
27▸              method ← "Gröbner basis method"                    ▷ Gröbner basis method
28▸          else
29▸              Break algorithm
30▸          end if
31▸      end if
32▸  end if
33▸ end procedure
```

# B  Original cases of R-matrices

This section is intended to demonstrate the number of solutions generated in the first steps of our procedure. After obtaining them, we distill the result list by employing $C \circ P$ conjugation, which refers to applying the $C$-transformation following the $P$ one. Further, we also employed various inversions $I(p_0 \to p_0^{-1})$ to redefine the parameters towards further simplification. We further collect them into the same equivalence classes by conjugating with the local basis transformation $K \times K$, where $K \in SL(2, C)$ (see section 2.2). Here, we also generate new constant solutions which are not present in Hietarinta's search. All the constants and functions $r_{1,2}(u)$ involved below are arbitrary. We also have an explicit form of parameters which transform these solutions into the one from the main text.

## B.1  $R$-matrices derived from rank 3 constant solutions

1.
$$\begin{pmatrix} r_1(u) & 0 & 0 & 0 \\ 0 & 0 & e^{c_1 u} r_1(u) & -(e^{c_1 u} - 1) r_1(u) \\ 0 & 0 & 0 & r_1(u) \\ 0 & 0 & 0 & r_1(u) \end{pmatrix}$$

2.
$$\begin{pmatrix} 0 & 0 & 0 & 0 \\ 0 & 0 & e^{c_2 u} r_1(u) & 0 \\ 0 & e^{c_1 u} r_1(u) & 0 & 0 \\ 0 & 0 & 0 & r_1(u) \end{pmatrix}$$

3.
$$\begin{pmatrix} 0 & 0 & r_1(u) & 0 \\ 0 & r_1(u) & 0 & 0 \\ 0 & 0 & r_1(u) & 0 \\ 0 & 0 & 0 & r_1(u) \end{pmatrix}$$

4.
$$\begin{pmatrix} r_1(u) & 0 & 0 & 0 \\ 0 & 0 & 0 & r_1(u) \\ 0 & 0 & r_1(u) & 0 \\ 0 & 0 & 0 & r_1(u) \end{pmatrix}$$

5.
$$\begin{pmatrix} r_1(u) & 0 & 0 & 0 \\ 0 & r_1(u) & 0 & 0 \\ 0 & 0 & \frac{q_0 r_1(u)}{p_0 + q_0} & \frac{q_0 r_1(u)}{p_0 + q_0} \\ 0 & 0 & \frac{p_0 r_1(u)}{p_0 + q_0} & \frac{p_0 r_1(u)}{p_0 + q_0} \end{pmatrix}$$

6.
$$\begin{pmatrix} 0 & -r_1(u) & r_1(u)\left(-e^{\frac{c_1 u}{q_0}}\right) & r_2(u) \\ 0 & 0 & -\frac{k_0 r_1(u) e^{\frac{c_1 u}{q_0}}}{q_0} & r_1(u) e^{\frac{c_1 u}{q_0}} \\ 0 & -\frac{k_0 r_1(u)}{q_0} & 0 & r_1(u) \\ 0 & 0 & 0 & 0 \end{pmatrix}$$

7.

$$
\begin{pmatrix}
0 & r_1(u) & r_1(u)e^{\frac{c_1 u}{p_0}} & r_2(u) \\
0 & 0 & \frac{k_0 r_1(u)e^{\frac{c_1 u}{p_0}}}{p_0} & \frac{q_0 r_1(u)e^{\frac{c_1 u}{p_0}}}{p_0} \\
0 & \frac{k_0 r_1(u)}{p_0} & 0 & \frac{q_0 r_1(u)}{p_0} \\
0 & 0 & 0 & 0
\end{pmatrix}
$$

8.

$$
\begin{pmatrix}
0 & r_1(u) & r_1(u)e^{\frac{c_1 u}{q_0}} & r_2(u) \\
0 & 0 & \frac{k_0 r_1(u)e^{\frac{c_1 u}{q_0}}}{q_0} & r_1(u)e^{\frac{c_1 u}{q_0}} \\
0 & \frac{k_0 r_1(u)}{q_0} & 0 & r_1(u) \\
0 & 0 & 0 & 0
\end{pmatrix}
$$

9.

$$
\begin{pmatrix}
0 & -r_1(u) & r_1(u)\left(-e^{\frac{c_1 u}{p_0}}\right) & r_2(u) \\
0 & 0 & \frac{k_0 r_1(u)e^{\frac{c_1 u}{p_0}}}{p_0} & r_1(u)e^{\frac{c_1 u}{p_0}} \\
0 & \frac{k_0 r_1(u)}{p_0} & 0 & r_1(u) \\
0 & 0 & 0 & 0
\end{pmatrix}
$$

10.

$$
\begin{pmatrix}
0 & r_1(u) & r_1(u)e^{\frac{c_1 u}{p_0}} & r_2(u) \\
0 & 0 & \frac{k_0 r_1(u)e^{\frac{c_1 u}{p_0}}}{p_0} & r_1(u)e^{\frac{c_1 u}{p_0}} \\
0 & \frac{k_0 r_1(u)}{p_0} & 0 & r_1(u) \\
0 & 0 & 0 & 0
\end{pmatrix}
$$

## B.2 Constant $R$-matrices which do not allow for a one-parameter Baxterization

1.

$$
\begin{pmatrix}
0 & 0 & 0 & 1 \\
0 & 0 & q_0 & 0 \\
0 & q_0 & 0 & 0 \\
q_0^2 & 0 & 0 & 0
\end{pmatrix}
$$

2.

$$
\begin{pmatrix}
1 & 0 & 0 & \frac{1}{2}-\frac{i}{2} \\
0 & 0 & \frac{1}{2}-\frac{i}{2} & 0 \\
0 & \frac{1}{2}-\frac{i}{2} & 0 & 0 \\
\frac{1}{2}-\frac{i}{2} & 0 & 0 & -i
\end{pmatrix}
$$

3.

$$
\begin{pmatrix}
1 & 0 & 0 & \frac{1}{2}+\frac{i}{2} \\
0 & 0 & \frac{1}{2}+\frac{i}{2} & 0 \\
0 & \frac{1}{2}+\frac{i}{2} & 0 & 0 \\
\frac{1}{2}+\frac{i}{2} & 0 & 0 & i
\end{pmatrix}
$$

4.

$$
\begin{pmatrix}
0 & 0 & 0 & -\frac{1}{\sqrt{2}} \\
0 & 1 & -\frac{1}{\sqrt{2}} & 0 \\
0 & -\frac{1}{\sqrt{2}} & 1 & 0 \\
-\frac{1}{\sqrt{2}} & 0 & 0 & -\sqrt{2}
\end{pmatrix}
$$

5.

$$\begin{pmatrix} 1 & 0 & 0 & 0 \\ 0 & -1 & 0 & 0 \\ 0 & 2 & 1 & 0 \\ 0 & 0 & 0 & 1 \end{pmatrix}$$

6.

$$\begin{pmatrix} 0 & 0 & 0 & \frac{1}{\sqrt{2}} \\ 0 & 1 & \frac{1}{\sqrt{2}} & 0 \\ 0 & \frac{1}{\sqrt{2}} & 1 & 0 \\ \frac{1}{\sqrt{2}} & 0 & 0 & \sqrt{2} \end{pmatrix}$$

7.

$$\begin{pmatrix} 1 & 0 & 0 & 0 \\ 0 & q_0 & 0 & 0 \\ 0 & 1-q_0 & 1 & 0 \\ 0 & 0 & 0 & -q_0 \end{pmatrix}$$

8.

$$\begin{pmatrix} 1 & 0 & 0 & \frac{q_0^2-1}{q_0^2-2q_0-1} \\ 0 & \frac{q_0^2+1}{-q_0^2+2q_0+1} & \frac{q_0^2-1}{q_0^2-2q_0-1} & 0 \\ 0 & \frac{q_0^2-1}{q_0^2-2q_0-1} & \frac{q_0^2+1}{-q_0^2+2q_0+1} & 0 \\ \frac{q_0^2-1}{q_0^2-2q_0-1} & 0 & 0 & \frac{q_0^2+2q_0-1}{q_0^2-2q_0-1} \end{pmatrix}$$

## B.3   New Baxterized $R$-matrices

1.

$$\begin{pmatrix} 1 & 0 & 0 & 0 \\ 0 & -1 & 0 & 0 \\ 0 & 0 & -1 & 0 \\ r_1(u) & 0 & 0 & -1 \end{pmatrix}$$

2.

$$\begin{pmatrix} 1 & 0 & 0 & 0 \\ 0 & -1 & 0 & 0 \\ 0 & 0 & -1 & 0 \\ r_1(u) & 0 & 0 & 1 \end{pmatrix}$$

3.

$$\begin{pmatrix} 1 & 0 & 0 & r_1(u) \\ 0 & -1 & q_0 r_1(u) & 0 \\ 0 & q_0 r_1(u) & -1 & 0 \\ q_0^2 r_1(u) & 0 & 0 & 1 \end{pmatrix}$$

4.

$$\begin{pmatrix} 0 & 0 & 0 & 1 \\ 0 & 0 & r_1(u) & 0 \\ 0 & r_1(u) & 0 & 0 \\ \frac{q_0}{p_0} & 0 & 0 & 0 \end{pmatrix}$$

5.

$$\begin{pmatrix} 0 & 1 & 0 & r_1(u) \\ q_0 & 0 & q_0 r_1(u) & 0 \\ 0 & q_0 r_1(u) & 0 & 1 \\ q_0{}^2 r_1(u) & 0 & q_0 & 0 \end{pmatrix}$$

6.

$$\begin{pmatrix} 1 & 0 & 0 & 0 \\ 0 & -i & 0 & 0 \\ 0 & 2e^{\frac{1}{2}(c_2-2c_1)u} & -i & 0 \\ 0 & 0 & 0 & 1 \end{pmatrix}$$

7.

$$\begin{pmatrix} 1 & 0 & 0 & 0 \\ 0 & i & 0 & 0 \\ 0 & 2e^{\frac{1}{2}(c_2-2c_1)u} & i & 0 \\ 0 & 0 & 0 & 1 \end{pmatrix}$$

8.

$$\begin{pmatrix} 1 & 0 & 0 & 0 \\ 0 & 1 & 0 & 0 \\ 0 & 0 & 1 & 0 \\ r_1(u) & 0 & 0 & -1 \end{pmatrix}$$

9.

$$\begin{pmatrix} 1 & 0 & 0 & 0 \\ 0 & -1 & 0 & 0 \\ 0 & 2e^{\frac{1}{2}(c_2-2c_1)u} & 1 & 0 \\ 0 & 0 & 0 & 1 \end{pmatrix}$$

10.

$$\begin{pmatrix} 1 & 0 & 0 & k_0 e^{\frac{c_2 u}{k_0}-c_1 u} \\ 0 & -1 & 0 & 0 \\ 0 & 2e^{\frac{c_2 u}{k_0}-c_1 u} & 1 & 0 \\ 0 & 0 & 0 & 1 \end{pmatrix}$$

11.

$$\begin{pmatrix} 1 & 0 & 0 & e^{(c_2-c_1)u} \\ 0 & -1 & e^{(c_2-c_1)u} & 0 \\ 0 & e^{(c_2-c_1)u} & 1 & 0 \\ -e^{(c_2-c_1)u} & 0 & 0 & 1 \end{pmatrix}$$

12.

$$\begin{pmatrix} 1 & 0 & 0 & 0 \\ 0 & 1 & 0 & 0 \\ 0 & 0 & 1 & 0 \\ r_1(u) & 0 & 0 & 1 \end{pmatrix}$$

13.

$$\begin{pmatrix} 1 & 0 & 0 & r_1(u) \\ 0 & 1 & q_0 r_1(u) & 0 \\ 0 & q_0 r_1(u) & 1 & 0 \\ q_0{}^2 r_1(u) & 0 & 0 & 1 \end{pmatrix}$$

14.

$$
\begin{pmatrix}
1 & r_1(u) & 0 & r_2(u) \\
q_0 r_1(u) & 1 & q_0 r_2(u) & 0 \\
0 & q_0 r_2(u) & 1 & r_1(u) \\
q_0{}^2 r_2(u) & 0 & q_0 r_1(u) & 1
\end{pmatrix}
$$

15.

$$
\begin{pmatrix}
1 & r_1(u) & 0 & 0 \\
c_1 r_1(u) & 1 & 0 & 0 \\
0 & 0 & 1 & r_1(u) \\
0 & 0 & c_1 r_1(u) & 1
\end{pmatrix}
$$

16.

$$
\begin{pmatrix}
1 & 0 & 0 & 0 \\
0 & 1 & 0 & 0 \\
0 & (p_0-1)\left(-e^{-u\left(\frac{c_2}{p_0-1}+c_1\right)}\right) & p_0 & 0 \\
0 & 0 & 0 & 1
\end{pmatrix}
$$

17.

$$
\begin{pmatrix}
1 & 0 & 0 & 0 \\
0 & q_0 & 0 & 0 \\
0 & (q_0-1)\left(-e^{-u\left(\frac{c_2}{q_0-1}+c_1\right)}\right) & 1 & 0 \\
0 & 0 & 0 & 1
\end{pmatrix}
$$

18.

$$
\begin{pmatrix}
1 & 0 & r_1(u) & 0 \\
0 & 1 & 0 & r_1(u) \\
0 & 0 & c_1 r_1(u)+1 & 0 \\
0 & 0 & 0 & c_1 r_1(u)+1
\end{pmatrix}
$$

19.

$$
\begin{pmatrix}
1 & 0 & r_1(u) & 0 \\
0 & 1 & 0 & r_1(u) \\
0 & 0 & c_1 r_1(u)+1 & 0 \\
0 & 0 & 0 & c_1 r_1(u)+1
\end{pmatrix}
$$

20.

$$
\begin{pmatrix}
1 & 0 & 0 & 0 \\
0 & 1 & 0 & 0 \\
r_1(u) & 0 & c_1 r_1(u)+1 & 0 \\
0 & r_1(u) & 0 & c_1 r_1(u)+1
\end{pmatrix}
$$

21.

$$
\begin{pmatrix}
1 & 0 & 0 & 0 \\
0 & q_0 & 0 & 0 \\
0 & (1-p_0 q_0)e^{-u\left(\frac{c_2}{p_0 q_0-1}+c_1\right)} & p_0 & 0 \\
0 & 0 & 0 & 1
\end{pmatrix}
$$

22.

$$
\begin{pmatrix}
1 & 0 & 0 & 0 \\
0 & q_0 & 0 & 0 \\
0 & (q_0-1)\left(-e^{-u\left(\frac{c_2}{q_0-1}+c_1\right)}\right) & 1 & 0 \\
0 & 0 & 0 & -q_0
\end{pmatrix}
$$

23.
$$\begin{pmatrix} 1 & 0 & 0 & 0 \\ 0 & q_0 & 0 & 0 \\ 0 & 2e^{\frac{1}{2}(c_2-2c_1)u} & -\frac{1}{q_0} & 0 \\ 0 & 0 & 0 & 1 \end{pmatrix}$$

24.
$$\begin{pmatrix} 1 & 0 & 0 & 0 \\ 0 & q_0 & 0 & 0 \\ 0 & (q_0+1)e^{u\left(\frac{c_2}{q_0+1}-c_1\right)} & -1 & 0 \\ 0 & 0 & 0 & 1 \end{pmatrix}$$

25.
$$\begin{pmatrix} 1 & 0 & 0 & 0 \\ 0 & q_0 & 0 & 0 \\ 0 & \left(q_0^2-1\right)\left(-e^{-u\left(\frac{c_2}{q_0^2-1}+c_1\right)}\right) & q_0 & 0 \\ 0 & 0 & 0 & 1 \end{pmatrix}$$

26.
$$\begin{pmatrix} 1 & 0 & 0 & 0 \\ 0 & q_0 & 0 & 0 \\ 0 & (1-p_0q_0)e^{-u\left(\frac{c_2}{p_0q_0-1}+c_1\right)} & p_0 & 0 \\ 0 & 0 & 0 & -p_0q_0 \end{pmatrix}$$

27.
$$\begin{pmatrix} 1 & r_1(u) & r_1(u) & 2c_1 r_1(u) \\ 0 & 1-\frac{r_1(u)}{c_1} & 0 & -r_1(u) \\ 0 & 0 & 1-\frac{r_1(u)}{c_1} & -r_1(u) \\ 0 & 0 & 0 & 1 \end{pmatrix}$$

28.
$$\begin{pmatrix} 1 & r_1(u) & 0 & 0 \\ 0 & c_1 r_1(u)+1 & 0 & 0 \\ 0 & 0 & 1 & r_1(u) \\ 0 & 0 & 0 & c_1 r_1(u)+1 \end{pmatrix}$$

29.
$$\begin{pmatrix} 1 & r_1(u) & r_2(u) & 0 \\ 0 & c_1 r_1(u)+1 & 0 & r_2(u) \\ 0 & 0 & c_1 r_2(u)+1 & r_1(u) \\ 0 & 0 & 0 & c_1\left(r_1(u)+r_2(u)\right)+1 \end{pmatrix}$$

30.
$$\begin{pmatrix} 1 & r_1(u) & r_1(u) & 2c_1 r_1(u) \\ 0 & 1-\frac{r_1(u)}{c_1} & 0 & -r_1(u) \\ 0 & 0 & 1-\frac{r_1(u)}{c_1} & -r_1(u) \\ 0 & 0 & 0 & 1 \end{pmatrix}$$

31.
$$\begin{pmatrix} 1 & r_1(u) & 0 & 0 \\ c_1 r_1(u) & c_2 r_1(u)+1 & 0 & 0 \\ 0 & 0 & 1 & r_1(u) \\ 0 & 0 & c_1 r_1(u) & c_2 r_1(u)+1 \end{pmatrix}$$

32.
$$\begin{pmatrix} 1 & r_1(u) & r_2(u) & 0 \\ 0 & c_1 r_1(u)+1 & 0 & r_2(u) \\ 0 & 0 & c_1 r_2(u)+1 & r_1(u) \\ 0 & 0 & 0 & c_1(r_1(u)+r_2(u))+1 \end{pmatrix}$$

33.
$$\begin{pmatrix} 1 & r_1(u) & r_1(u) & 2c_1 r_1(u) \\ 0 & 1-\frac{r_1(u)}{c_1} & 0 & -r_1(u) \\ 0 & 0 & 1-\frac{r_1(u)}{c_1} & -r_1(u) \\ 0 & 0 & 0 & 1 \end{pmatrix}$$

34.
$$\begin{pmatrix} 1 & r_1(u) & r_1(u) & 2c_2 r_1(u) \\ c_1 r_1(u) & 1-\frac{r_1(u)}{c_2} & 2c_1 c_2 r_1(u) & -r_1(u) \\ c_1 r_1(u) & 2c_1 c_2 r_1(u) & 1-\frac{r_1(u)}{c_2} & -r_1(u) \\ 2c_1^2 c_2 r_1(u) & -c_1 r_1(u) & -c_1 r_1(u) & 1 \end{pmatrix}$$

35.
$$\begin{pmatrix} 1 & r_1(u) & r_2(u) & 0 \\ 0 & c_1 r_1(u)+1 & 0 & r_2(u) \\ 0 & 0 & c_1 r_2(u)+1 & r_1(u) \\ 0 & 0 & 0 & c_1(r_1(u)+r_2(u))+1 \end{pmatrix}$$

36.
$$\begin{pmatrix} 1 & 0 & 0 & 0 \\ r_1(u) & 1-\frac{r_1(u)}{c_1} & 0 & 0 \\ r_1(u) & 0 & 1-\frac{r_1(u)}{c_1} & 0 \\ 2c_1 r_1(u) & -r_1(u) & -r_1(u) & 1 \end{pmatrix}$$

37.
$$\begin{pmatrix} 1 & 0 & 0 & 0 \\ r_1(u) & c_1 r_1(u)+1 & 0 & 0 \\ 0 & 0 & 1 & 0 \\ 0 & 0 & r_1(u) & c_1 r_1(u)+1 \end{pmatrix}$$

38.
$$\begin{pmatrix} 1 & 0 & 0 & 0 \\ r_1(u) & c_1 r_1(u)+1 & 0 & 0 \\ r_2(u) & 0 & c_1 r_2(u)+1 & 0 \\ 0 & r_2(u) & r_1(u) & c_1(r_1(u)+r_2(u))+1 \end{pmatrix}$$

39.
$$\begin{pmatrix} 1 & 0 & r_1(u) & 0 \\ 0 & 1 & 0 & r_1(u) \\ c_1 r_1(u) & 0 & 1 & 0 \\ 0 & c_1 r_1(u) & 0 & 1 \end{pmatrix}$$

40.
$$\begin{pmatrix} 1 & 0 & r_1(u) & r_2(u) \\ 0 & 1 & c_1 r_2(u) & r_1(u) \\ c_1 r_1(u) & c_1 r_2(u) & 1 & 0 \\ c_1^2 r_2(u) & c_1 r_1(u) & 0 & 1 \end{pmatrix}$$

41.

$$\begin{pmatrix} 1 & 0 & r_1(u) & 0 \\ 0 & 1 & 0 & r_1(u) \\ c_1 r_1(u) & 0 & c_2 r_1(u) + 1 & 0 \\ 0 & c_1 r_1(u) & 0 & c_2 r_1(u) + 1 \end{pmatrix}$$

42.

$$\begin{pmatrix} 1 & r_1(u) & r_2(u) & 0 \\ c_1 r_1(u) & c_2 r_1(u) + 1 & 0 & r_2(u) \\ c_1 r_2(u) & 0 & c_2 r_2(u) + 1 & r_1(u) \\ 0 & c_1 r_2(u) & c_1 r_1(u) & c_2\left(r_1(u) + r_2(u)\right) + 1 \end{pmatrix}$$

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
