# Peer review of "New spectral-parameter dependent solutions of the Yang-Baxter equation"

_SciPost Physics_

## Round 2 · Referee Report · Anonymous (Referee 1) · 2024-5-5

Strengths

1. It is innovative and solves a relevant and difficult problem in the field.
2. It is well written.

Weaknesses

N/a

Report

This paper addresses the difficult question of finding new solutions of the Yang-Baxter equation (YBE), even when no specific symmetries are assumed. It applies in particular to classifying non-regular R-matrices. They created a systematic procedure to solve the YBE, with  a detailed algorithm that can be implemented in Mathematica. 
In this paper, in addition to presenting their procedure, they applied it to classify non-regular 4x4 R-matrices both for full rank and rank-3 cases, finding several new integrable models.
There are few points that could be clarified and some typos to be corrected.  But in general the paper is well structured and easy to read. It addresses a relevant and difficult problem in the field. The procedure introduced in the paper is novel and it has the potential to be applied to several other cases, and even for other equations related to the YBE, like the boundary YBE and the RLL equations. Therefore, I recommend the paper to publication in SciPost Physics after the points below are addressed.

Requested changes

Points in sections A, B and C in the report.pdf file attached below

Attachment

Recommendation

Ask for minor revision

---

## Round 2 · Referee Report · Anonymous (Referee 2) · 2024-5-10

Strengths

Novel algebro-differential tool with autonomous properties for the search of new integrable models. Should be applicable to any symmetry algebra, dimensionality, no dependence on model characteristic physical properties.

Weaknesses

No particular novelty for the two-dimensional case, some of the apparatus steps might need resolution for higher-dim setups.

Language/Errata and related in the article must be reviewed.

Report

The authors address a technical tool for finding and classifying the solution space of Yang-Baxter equation for the case of 2-dim local quantum space. It is based on composing algebro-differential conditions following the quantum Yang-Baxter Equation along with its invariance properties.

1). In particular, it is not clear of how much novelty comes from the found solutions itself, since 2-dim case has been already classified under generic conditions in several works, including 1712.02341 up to 8-vertex and 1904.12005, where generic 16-vertex prescription with C2 space provided the full set of generating solutions. From the latter work, one can reproduce any 2-dim solution (incl. graded ones in C1|1, explored by Sklyanin and Takhtajan) by applying YBE invariant transformations and appropriate parametric (or functional) identifications. The solution generators of 1904.12005 have been also recently confirmed in 2304.07247 by exploiting neural network architecture. Moreover in AdS spin chain picture (2003.04332, 2109.00017), the new models with arbitrary spectral dependence have been foundas, which clearly includes the C2 solution space of the (additive) YBE.

"The main idea behind our proposal is to solve YB relations step-by-step in a manner that we avoid obtaining repeated solutions, which we encounter while resolving an over-determined system of equations. " … "There are two different means of resolution for the relations: First to solve the differential ones and secondly to consider them as system of polynomials that are reducible through Gröbner basis calculations. " … "The red dots indicate terminated vertices which may indicate a valid solution or an invalid (often indeterminate) one. "

- Is it sufficient, in the given prescription, to consider a vertex to represent an invalid solution if it was marked as "ïndeterminate"? (since in certain cases (incl. higher dimensional) the final constraints may require an appropriate regulation to lead to a consistent solution)

- It is also important not only to reduce degeneracy of the solution space in the arising overdetermined system, but once it is acquired to map various solutions under established transformations and identify associated subclasses - that would form characteristic generators, which will cover the full solution space. In addition, if one uses allowed rescaling/fixings from the beginning, then the constraint space at the start would drop accordingly. In this respect, the number in B.1, B.2 and B.3 should be possible to reduce? (hence solutions analogous to 34. and 42. after stated operations should be mapped to the ones found in the works above)

2). This work appears interesting from the technical perspective, which constitutes an autonomous algorithmic search for new integrable models, in this case stemming from the YBE. It is somewhat unobvious whether it is a lot more advantageous (time/complexity/intermediate analytic structure) over the approaches implemented in the series of works above, because nonlinearity and algebraic complexity can become subtle (even in the intermediate steps), especially if one lifts to higher D (incl. superalgebras). Nevertheless, in a number of cases it might appear very useful to recast integrable data into coupled differential and algebraic constraints directly following from YBE and attempt iteratively.

In this regard, it would be important to check implementation of the present approach to integrable systems with higher-dim local spaces, since raising dof-complexity in many cases leads to mid-step constraints that require separate treatment to be solved (in fact, such issues have been observed through automated resolution for the systems with 4-dim local spaces and generic spectral dependence, e.g. Hubbard-Shastry type class on AdS_5 background).

"An notable feature of this process is that all terminated search branches from the final output of the program can be re-executed with broader determinants. Hence one can extend the incomplete tedious algorithmic search in the future." … " nonlinear relations (those with power terms) explicitly and considered simplifying them through Gröbner computations. "

- Might the current method be free of its autonomy failure for higher dimensional local spaces? (Also Baxterisation or linearisation with Gröbner reductions can become inapplicable)

- Can it be applied to d-simplexes (Tetrahedron and higher)?

Apart from the stated above, it would relevant to correct/improve the following parts of the text, which appear a bit obscure (along with multiple typos):

p. 6: "... set the spectral parameter u towards zero, leading to the below relations" (ex. u0 leads to …)

p. 6: "all enlisted invertible constant solutions of the YBE of the lowest
dimensions, which we refer them into classes" ; "Each class denote
to a initial condition for solving the YB relations" ;

p. 12: "Relations in D(u) and D(0), which has no differential form is shifted in A(u)."

p. 12: "... algorithm deals with resolving A(u) and D(u) which we describe the following sections."

p. 12: "There are two different means of resolution for the relations …"

p. 12: "... to control the complexity of the obtained solutions even after solving them, …"

p. 13: "After then by employing the symmetries …"

Errata:

p. 5: "These relations from Eqs. (10) and (11) only contains the spectral parameter u." (-> contain only/depend solely on the spectral parameter u)

p. 12: "Each of them then represent search branches …" (->represents)

p. 13: "Any residual case where it does not resolve the YBE is solved in Stage 4." (-> … , which does not resolve the YBE, is solved in Stage 4.)

p. 13: "The ones which solves … " (-> solve)

p. 15: "An notable feature of this process …" (-> A …)

Comment: I recommend authors to review errata and related in the article. Conventionally it could be useful to enclose an ancillary Mathematica notebook with examples (instead/along with pseudocodes).

Recommendation

Ask for major revision

---

## Editorial Decision

awaiting_resubmission